# Association of Specific Comorbidities with Monosodium Urate Crystal Deposition in Urate-Lowering Therapy-Naive Gout Patients: A Cross-Sectional Dual-Energy Computed Tomography Study

**DOI:** 10.3390/jcm9051295

**Published:** 2020-05-01

**Authors:** Tristan Pascart, André Ramon, Sébastien Ottaviani, Julie Legrand, Vincent Ducoulombier, Eric Houvenagel, Laurène Norberciak, Pascal Richette, Fabio Becce, Paul Ornetti, Jean-François Budzik

**Affiliations:** 1Department of Rheumatology, Lille Catholic Hospitals, University of Lille, F-59160 Lomme, France; ducoulombier.vincent@ghicl.net (V.D.); houvenagel.eric@ghicl.net (E.H.); 2EA 4490, PMOI, Physiopathologie des Maladies Osseuses Inflammatoires, University of Lille, F-59000 Lille, France; budzik.jean-francois@ghicl.net; 3Department of Rheumatology, Dijon University Hospital, University of Bourgogne, F-21000 Dijon, France; andre.ramon@chu-dijon.fr (A.R.); paul.ornetti@chu-dijon.fr (P.O.); 4Department of Rheumatology, Hôpital Bichat, AP-HP, F-75018 Paris, France; sebastien.ottaviani@aphp.fr; 5Department of Diagnostic and Interventional Radiology, Lille Catholic Hospitals, University of Lille, F-59160 Lomme, France; legrand.julie@ghicl.net; 6Department of Medical Research, Biostatistics, Lille Catholic Hospitals, University of Lille, F-59160 Lomme, France; norberciak.laurène@ghicl.net; 7Department of Rheumatology, Hôpital Lariboisière, AP-HP, F-75010 Paris, France; pascal.richette@aphp.fr; 8INSERM U1132, Université Paris Diderot, F-75010 Paris, France; 9Department of Diagnostic and Interventional Radiology, Lausanne University Hospital and University of Lausanne, 1011 Lausanne, Switzerland; fabio.becce@chuv.ch

**Keywords:** chronic heart failure, comorbidities, diabetes mellitus, dual-energy computed tomography, gout, monosodium urate crystal deposition

## Abstract

(1) Background: To determine which factors are associated with the volume of monosodium urate (MSU) crystal deposition quantified by dual-energy computed tomography (DECT) in urate-lowering therapy (ULT)-naive gout patients. (2) Methods: In this multicenter cross-sectional study, DECT scans of knees and feet/ankles were prospectively obtained from ULT-naive gout patients. Demographic, clinical (including gout history and comorbidities), and biological data were collected, and their association with DECT MSU crystal volume was analyzed using bivariate and multivariate analyses. A second bivariate analysis was performed by splitting the dataset depending on an arbitrary threshold of DECT MSU volume (1 cm^3^). (3) Results: A total of 91 patients were included. In the bivariate analysis, age (*p* = 0.03), gout duration (*p* = 0.003), subcutaneous tophi (*p* = 0.004), hypertension (*p* = 0.02), diabetes mellitus (*p* = 0.05), and chronic heart failure (*p* = 0.03) were associated with the total DECT volume of MSU crystal deposition. In the multivariate analysis, factors associated with DECT MSU volumes ≥1 cm^3^ were gout duration (odds ratio (OR) for each 10-year increase 3.15 (1.60; 7.63)), diabetes mellitus (OR 4.75 (1.58; 15.63)), and chronic heart failure (OR 7.82 (2.29; 31.38)). (4) Conclusion: Specific comorbidities, particularly chronic heart failure and diabetes mellitus, are more strongly associated with increased MSU crystal deposition in knees and feet/ankles than gout duration, regardless of serum urate level.

## 1. Introduction

Gout causes recurrent flares and chronic gouty arthritis induced by monosodium urate (MSU) crystal deposition secondary to long-standing hyperuricemia [1,2]. The extent of MSU crystal deposition in gout patients appears to determine disease activity [3], and tophaceous gout is associated with an increased mortality risk [4]. Several factors, in particular cardiovascular comorbidities, have been associated with extensive MSU deposition in the subset of tophaceous gout patients, who represent only 10% to 20% of the gout patient population overall [4,5].

Dual-energy computed tomography (DECT) is an increasingly available imaging technique that allows for a more rapid, detailed, and accurate assessment of the burden of MSU crystal deposition than clinical evaluation of subcutaneous tophi, by providing direct quantification and mapping of MSU volume in soft tissues [6,7]. DECT is currently mainly limited by spatial resolution (minimum ~250 µm in-plane) and certain typical artifacts, which are, however, easily recognized and dealt with [8,9]. While MSU crystal formation is enhanced by certain factors in vitro, such as pH, temperature, and other ion concentrations [10], it is currently unknown whether specific comorbidities and clinical features are associated with increased MSU deposition in vivo. Furthermore, no study has yet evaluated the MSU crystal volume in urate-lowering therapy (ULT)-naive patients.

Therefore, the aim of our study was to determine which factors are associated with the volume of MSU crystal deposition quantified by DECT of the knees and feet/ankles in ULT-naive gout patients.

## 2. Experimental Section

In this multicenter cross-sectional study, DECT scans of the knees and feet/ankles were prospectively obtained from February 2016 to November 2019 in patients diagnosed with gout according to the 2015 ACR/EULAR gout classification criteria [11] and who were ULT-naive, in Lille Catholic University Hospitals, Dijon University Hospital, and Bichat-Paris University Hospital (France). The study was approved by the institutional review board of Lille Catholic University Hospitals (protocol 2016-04-16).

Consecutive patients seen during their first medical consultations in one of three French tertiary centers with a diagnosis of gout [12] and no recollection of having taken ULTs previously were enrolled to undergo DECT scans for evaluation and quantification of their MSU crystal volume. Demographic, clinical (including gout history and comorbidities), and biological data were collected during this first clinical visit. Serum urate levels [12] were measured in the inter-critical period [13]. Measuring lipid levels was not mandatory as per study design. The following comorbidities were considered present if a prior diagnosis had been established by their treating physician: hypertension, diabetes mellitus, chronic heart failure, urolithiasis, stroke, myocardial infarction.

DECT scans of the knees and feet/ankles were performed using single-source DECT systems (Somatom Definition Edge; Siemens Healthineers, Erlangen, Germany; or Toshiba Aquilion One Genesis, Canon Medical, Tokyo, Japan). Details of the standardized DECT protocols and excellent reliability of the automated post-processing and quantification of MSU crystal volumes deposited in the knees and feet/ankles, with manual removal of typical artifacts by experienced musculoskeletal radiologists, are described elsewhere [8,9,14]. In all three different centers, DECT images were reconstructed at a section thickness/interval of approximately 0.6/0.3 mm, yielding almost isotropic voxels of approximately 0.6 × 0.6 × 0.6 mm. DECT volume measurements were performed routinely using the default gout post-processing algorithms from each CT vendor.

Statistical analyses were performed using R software (version 3.4.2, R Foundation for Statistical Computing, Vienna, Austria). Means ± standard deviations (SD), supplemented with medians (interquartile ranges) where appropriate, and percentages were used to describe clinical and biological patient characteristics. No imputations were performed for missing data. The association between DECT MSU volumes and all other quantitative data were first evaluated using Spearman’s correlation coefficient, while qualitative data were assessed with the Wilcoxon-Mann-Whitney test. A multivariate analysis was then applied to search for factors affecting MSU volumes using multiple linear regression models integrating variables with *p*-values < 0.2 in bivariate analysis. As residuals were not normally distributed, adjusted R-squared was used for the reduced model, and 95% confidence intervals (CI) of coefficients were assessed by bootstrapping.

In addition, a second bivariate analysis was performed by splitting the dataset depending on an arbitrary threshold of DECT MSU volume (1 cm^3^), which was considered clinically relevant for predicting the risk of gout flares [3]. Student’s t or Wilcoxon-Mann-Whitney tests and Chi-squared or Fisher’s exact tests were used, where appropriate. A second multivariate analysis using binary logistic regression models was built with variables exhibiting *p* < 0.2 in the bivariate analysis. The cut-off of *p* < 0.2 was specifically chosen to avoid missing variables, which could become statistically significant when adjusted on the other factors in the multivariate analysis (rather than taking variables with *p* < 0.05 in bivariate analysis). The step-by-step backward method, based on the Akaike information criterion, was selected for the reduced model. Validation and reduced model performance were assessed using the area under the receiver operating characteristic (ROC) curve. The best-discriminating cut-off value (Youden index) was derived from the ROC curve. The significance level was set at *p* < 0.05.

## 3. Results

### 3.1. Patient Characteristics

A total of 125 ULT-naive gout patients were included in this cohort, of whom 91 patients underwent both DECT scans of the knees and feet/ankles (Figure 1).

Table 1 reports the clinical and biological features of patients from the study population included in the analysis (*n* = 91).

The DECT volume of MSU crystal deposition in the knees and feet/ankles was 3.5 ± 11.5 cm^3^ (0.54 (0.16; 2.26) cm^3^). The DECT volume of MSU crystal deposition was significantly higher in the feet/ankles (2.2 ± 8.6 cm^3^ (0.32 (0.08; 1.45) cm^3^)) than in the knees (1.3 ± 3.9 cm^3^ (0.13 (0.03; 0.57) cm^3^)) (*p* = 0.006), and these volumes were moderately correlated (*r* = 0.53 (0.34; 0.67)). Patients with early gout (less than two years from gout onset) presented with average DECT MSU volumes of 0.55 ± 0.81 and 0.36 ± 0.69 cm^3^ in the feet/ankles and knees, respectively, compared with 4.0 ± 12.7 (*p* = 0.02) and 1.9 ± 4.7 cm^3^ (*p* = 0.12) when gout lasted for more than two years.

### 3.2. Association Between Factors and the Total DECT Volume of MSU Crystal Deposition in the Knees and Feet/Ankles

After the bivariate analysis, several specific factors were significantly associated with the total DECT volume of MSU crystal deposition in the knees and feet/ankles: age (*p* = 0.03), gout duration (*p* = 0.003), presence of subcutaneous tophi (*p* = 0.004), hypertension (*p* = 0.02), diabetes mellitus (*p* = 0.05), and chronic heart failure (CHF) (*p* = 0.03) (Table 2 and Table 3).

The following factors were ultimately included in the reduced model: age (coefficient −0.15 (−0.36; 0.05)), gout duration (coefficient 0.36 (0.02; 0.97)), number of gout flares in last six months (coefficient 0.86 (−0.43; 3.45)), presence of subcutaneous tophi (coefficient 7.3 (−0.21; 16.93), and CHF (coefficient 7.13 (−2.32; 16.36)) (Table 4).

The following factors were not retained by automatic selection in the reduced model (and did not demonstrate a significant association with MSU crystal DECT volume in multivariate analysis): eGFR, hypertension, history of myocardial infarction, and diabetes mellitus. This model explained only 21% of the variance of MSU crystal deposition, with a residual SD of 10.41 cm^3^. The only variable significantly associated with MSU DECT volume was gout duration (Spearman correlation coefficient 0.36 (0.02; 0.97)), while subcutaneous tophi and CHF, despite high correlation coefficients, did not reach statistical significance as their 95% confidence intervals included 0.

### 3.3. Factors Associated with the Total DECT Volume of MSU Crystal Deposition ≥1 cm^3^ in the Knees and Feet/Ankles

After the bivariate analysis, the following factors were significantly associated with MSU DECT volume ≥ 1 cm^3^ in the knees and feet/ankles: age (*p* = 0.006), gout duration (*p* = 0.002), presence of subcutaneous tophi (*p* = 0.02), hypertension (*p* < 0.001), and CHF (*p* = 0.005) (Table 5).

Variables ultimately included in the reduced model and significantly associated with MSU DECT volumes ≥1 cm^3^ were: gout duration (OR for each 10-year increase in disease duration 3.15 (1.60; 7.63)), diabetes mellitus (OR 4.75 (1.58; 15.63)), and CHF (OR 7.82 (2.29; 31.38)) (Table 6).

Hypertension could not be included in this model because too few patients with high MSU volumes did not have hypertension (<10%), while subcutaneous tophi were not retained by the automatic statistical selection. The model performance was good, with an AUC of 0.816 (Figure 2). The best-case scenario of the model predicted a volume of MSU crystal deposition >1cm^3^ at the knees and feet/ankles with a sensitivity of 68.6% (95% CI (54.3; 82.9)) and a specificity of 78.9% (95% CI (67.3; 88.5)).

## 4. Discussion

This cross-sectional multicenter DECT study provides the first in-depth assessment of the relationship between several associated factors occurring during the natural history of ULT-naive gout patients and the volume of MSU crystal deposition quantified by DECT. The study suggests that several specific comorbidities, in particular CHF, diabetes mellitus, and hypertension, may play a predominant role in the extent of MSU crystal deposition in the knees and feet/ankles comparable to prolonged gout duration, regardless of serum urate level.

In our study, gout duration was not as important as expected to explain MSU crystallization in vivo. Subcutaneous tophi were expectedly associated with MSU crystal DECT volumes as palpable tophi mirror per se substantial MSU deposition in soft tissues. We found that tophaceous gout patients exhibited, on average, an almost 7-fold higher MSU DECT volume than non-tophaceous gout patients in multiple linear regression models (Table 4). Patient age and gout duration were expected to play important roles in the formation of the MSU crystal burden. The few studies that have investigated MSU deposition with DECT in subjects not receiving ULT focused only on asymptomatic hyperuricemia [15,16]. These studies reported that increasing age, and therefore likely hyperuricemia duration, was associated with silent MSU crystal deposition. However, this association, although being the only significant variable in both models, was not as strong as the presence of specific comorbid conditions in our study, suggesting that other coexisting factors than time alone are involved in the crystallization rate of the MSU burden. To illustrate this point, in the logistic regression model, patients with CHF had the same odds of having MSU crystal deposition >1 cm^3^ as patients with more than 20-year gout duration. In addition, serum urate levels had a poor correlation with MSU crystal DECT volumes in our study. This finding was surprising given that patients with subcutaneous tophi were reported to have higher serum urate levels than non-tophaceous gout patients [17]. This could suggest that as long as serum urate levels are above urate saturation levels, the kinetics of MSU crystallization are guided by comorbidities and time, except for patients with large macroscopic deposition in whom higher serum urate levels could play an additional role. This suggests that specific comorbidities are contributors to increased soft-tissue MSU deposition, regardless of serum urate level. Several comorbidities such as diabetes mellitus, high blood pressure, and CHF are commonly associated with hyperuricemia and gout [1,18]. In our study, these coexisting conditions were strongly associated with MSU crystal DECT volume, regardless of serum urate level, thereby suggesting an additional role in promoting MSU crystallization than that of hyperuricemia alone. Part of the explanation for why CHF is associated with higher MSU deposits could be the reduced velocity of peripheral blood flow, which could facilitate the crystallization process [19,20]. Sodium retention occurring during CHF could also contribute since sodium reduces urate solubility, which could also be itself associated with hypertension [10]. Conversely, while hypertension and endothelial dysfunction have been related to asymptomatic hyperuricemia itself, particularly in the higher quartiles of serum urate levels, and considering the increasing evidence of vascular MSU deposition in gout patients, MSU crystal formation and deposition could be the missing link between hyperuricemia and cardiovascular diseases in gout [21,22,23]. On the other hand, the association between gout and diabetes mellitus is known, as gout patients are at higher risk of developing diabetes [24]. However, the relationship between the two diseases is still unclear, and the strong association between MSU crystal volume and diabetes in our study suggests that MSU crystals may play a role, possibly through continuous crystal-induced inflammation. The poor performance of multiple linear regression models, which could explain only 21% of the variance of DECT volumes of MSU crystal deposition, suggests that additional causative factors have yet to be determined. Future research on genetics integrating DECT assessment of MSU crystal deposition will hopefully help determine which are these still unknown factors guiding MSU crystallization in vivo.

We acknowledge the following study limitations. First, the inherent limitation of a cross-sectional methodology implies that this study demonstrates associations but not causality. The association between CHF and MSU crystal DECT volume can illustrate both an increased crystallization process in the course of CHF, but also an increased incidence of CHF in patients with high MSU crystal volumes. Future prospective studies are therefore needed to determine more confidently the direction of the potential causal relationship between comorbidities and crystal deposition volume. Second, lipid levels are missing for a substantial number of patients, and the study may be underpowered to assess dyslipidemia as a potential other associated factor. Third, the choice of a volume of 1 cm^3^ as the threshold for clinically meaningful deposition relies on its predictive value for gout flares and may be considered arbitrary [3], and other volumes may be more relevant when considering comorbidities, but no other volumes have been suggested to be more relevant so far. This threshold did, however, separate the cohort well, into two even patient groups. Fourth, hypertension was considered to be associated with high DECT volumes of MSU crystal deposition, although the variable was not included in the multivariate models. The reason why it could not be included in these models is that hypertension was almost systematically present in patients with >1 cm^3^ of MSU crystal deposits, supporting the notion that this association truly exists, although the force of the relationship could not be assessed. Finally, MSU crystal volume measurements could vary to some extent depending on the type of DECT scanner, and post-processing protocol used. To our knowledge, comparative studies on the diagnostic accuracy of different DECT scanners in gout are not yet available, but MSU volumes are not expected to vary significantly if certain conditions are met. Moreover, for all DECT scanners involved in this study, we have used standardized protocols with default gout dual-energy post-processing algorithms, which have been used and reported in several publications with these two DECT manufacturers (Siemens Healthineers and Canon Medical) so far.

Given the important role played by the MSU crystal burden in the causal relationship between gout and its comorbidities, this study should encourage the identification of patients with high MSU crystal deposits, who are at higher risk of gout flares and mortality from cardiovascular diseases [4]. This study suggests that the MSU crystal volume may be the missing link in the relationship between gout and cardiovascular diseases.

## Figures and Tables

**Figure 1 jcm-09-01295-f001:**
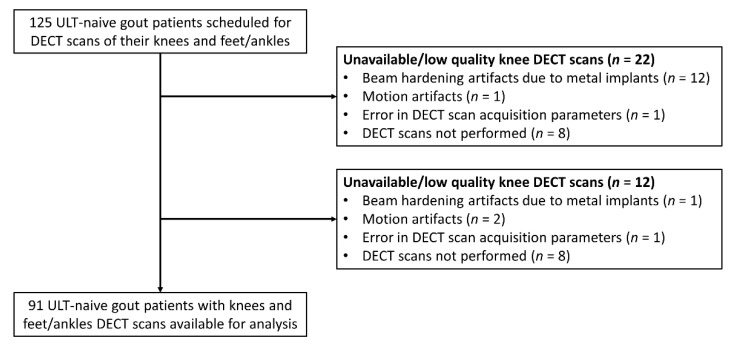
Study flow chart. ULT: urate lowering therapy; DECT: dual-energy computed tomography.

**Figure 2 jcm-09-01295-f002:**
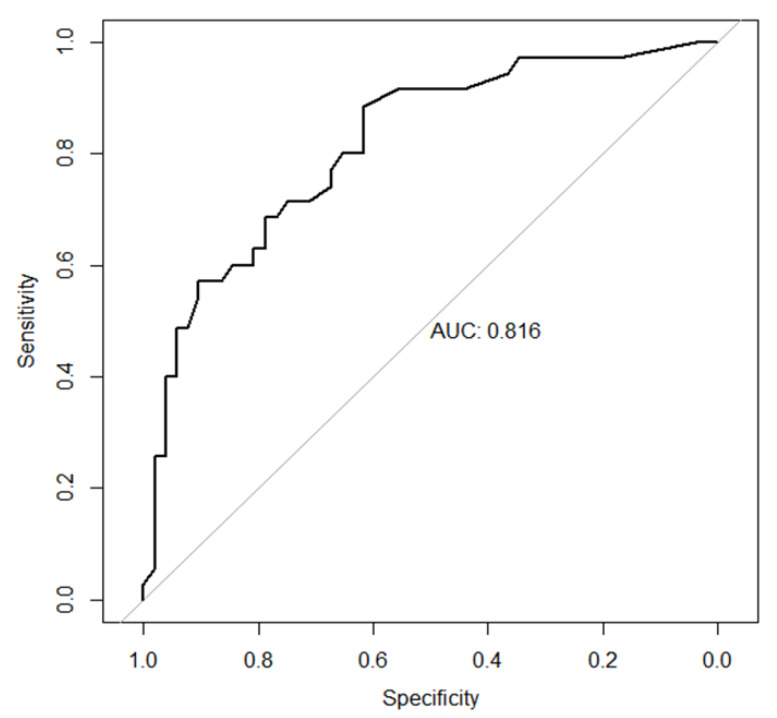
ROC curve of the model explaining DECT volumes of monosodium urate crystal ≥ 1 cm^3^ at the knees and feet/ankles.

**Table 1 jcm-09-01295-t001:** Patient characteristics.

	Patients (*n* = 91) Mean ± SD (Median (Interquartile Range))	Complete Data
Demographics		
Age (years)	65.3 ± 15.7	91
Gout duration (years)	7.4 ± 9.7 (3 (0.5; 8))	91
Male gender	76 (83.5%)	91
BMI (kg/m^2^)	29.2 ± 5.1	90
Disease history		
Number of flares (in last six months)	2.2 ± 2.2 (2 (1; 2))	90
Tophi	18 (20%)	90
Urolithiasis	13 (14.3%)	91
Comorbidities		
Hypertension	63 (70%)	90
Stroke	4 (4.5%)	89
Myocardial infarction	16 (18%)	89
Diabetes mellitus	26 (29.2%)	89
Chronic heart failure	18 (20.2%)	89
Ongoing drugs		
Diuretics	27 (30.3%)	89
Anti-inflammatory drugs	4 (4.4%)	90
Lipid-lowering drugs	35 (41.7%)	84
Laboratory results		
Serum urate level (mg/dL)	9.1 ± 2.2 (88 (77; 99))	88
eGFR (mL/min/1.73 m^2^)	70.7 ± 27.0	89
Triglycerides level (mg/dL)	197 ± 349 (160 (111; 206))	54
Cholesterol level (mg/dL)	179 ± 55	56
LDL level (mg/dL)	100 ± 39	54
HDL level (mg/dL)	43 ± 12	54

BMI: body mass index; eGFR: estimated glomerular filtration rate (CKD-EPI); DECT: dual-energy computed tomography; MSU: monosodium urate; LDL: low-density lipoproteins; HDL: high-density lipoproteins.

**Table 2 jcm-09-01295-t002:** Bivariate analysis of quantitative variables associated with monosodium urate crystal DECT volume at the knees and feet/ankles.

Variable	N	Spearman Correlation Coefficient	95% CI	*p*-Value
Age	91	0.22	(0.01; 0.42)	**0.03**
Gout duration	91	0.31	(0.10; 0.51)	**0.003**
Number of flares (in last six months)	90	0.19	(−0.03; 0.38)	0.08
eGFR	89	−0.14	(−0.34; 0.08)	0.19
Serum urate level	88	0.07	(−0.14; 0.26)	0.52
BMI	90	−0.10	(−0.32; 0.12)	0.37
Triglycerides level	53	0.04	(−0.26; 0.32)	0.77
Cholesterol level	56	−0.06	(−0.31; 0.20)	0.68
LDL level	54	−0.09	(−0.37; 0.20)	0.51
HDL level	54	0.06	(−0.21; 0.32)	0.66

BMI: body mass index; eGFR: estimated glomerular filtration rate (CKD-EPI); CI: confidence interval. Bold: significant *p*-values (*p* < 0.05).

**Table 3 jcm-09-01295-t003:** Bivariate analysis of qualitative variables associated with monosodium urate crystal DECT volume at the knees and feet/ankles.

9		N	Median MSU Crystal DECT Volume (cm^3^) (Q1; Q3)	*p*-Value
Gender	Female	15	0.87 (0.18; 2.41)	0.70
	Male	76	0.45 (0.15; 2.08)	
Ongoing anti-inflammatory drugs	No	86	0.56 (0.16; 2.24)	0.48
	Yes	4	1.9 (0.29; 8.63)	
Subcutaneous tophi	No	72	0.43 (0.12; 1.90)	**0.004**
	Yes	18	2.68 (0.71; 6.33)	
Urolithiasis	No	78	0.56 (0.16; 2.24)	0.56
	Yes	13	0.3 (0.07; 2.28)	
Hypertension	No	27	0.38 (0.10; 0.62)	**0.02**
	Yes	63	1.01 (0.18; 2.66)	
Stroke	No	85	0.59 (0.16; 2.28)	0.06
	Yes	4	0.16 (0.12; 0.18)	
Myocardial infarction	No	73	0.46 (0.16; 1.96)	0.15
	Yes	16	2.05 (0.30; 4.46)	
Diabetes mellitus	No	63	0.41 (0.09; 2.11)	**0.05**
	Yes	26	1.09 (0.29; 2.63)	
Ongoing lipid-lowering drugs	No	49	0.54 (0.13; 2.28)	0.57
	Yes	35	0.36 (0.14; 2.11)	
Chronic heart failure	No	71	0.42 (0.12; 1.96)	**0.03**
	Yes	18	2.04 (0.70; 2.95)	
Ongoing diuretics	No	62	0.45 (0.16; 2.26)	0.89
	Yes	27	1.79 (0.13; 2.26)	
Gout duration	≤2 years	37	0.25 (0.10; 0.70)	**0.007**
	>2 years	54	1.01 (0.22; 3.00)	
eGFR (mL/min/1.73 m^2^)	<60	30	1.21 (0.19; 2.48)	0.51
	≥60	59	0.44 (0.12; 1.99)	

eGFR: estimated glomerular filtration rate (CKD-EPI); DECT: dual-energy computed tomography; MSU: monosodium urate. Bold: significant *p*-values (*p* < 0.05).

**Table 4 jcm-09-01295-t004:** Multivariate analysis (linear regression model) of factors explaining monosodium urate crystal DECT volume at the knees and feet/ankles.

Factor	Original Coefficient	Standard Error	95% CI
Age	−0.15	0.08	(−0.36; 0.05)
Gout duration	0.36	0.13	(0.02; 0.97)
Number of flares (in last six months)	0.86	0.51	(−0.43; 3.45)
Subcutaneous tophi	7.30	3.02	(−0.21; 16.93)
Chronic heart failure	7.13	2.99	(−2.32; 16.36)

Adjusted *R*² = 0.21; *F* = 5.6; *p* = 0.0002; Standard deviation of residuals: 10.41; CI: confidence interval.

**Table 5 jcm-09-01295-t005:** Factors associated in bivariate analysis with monosodium urate (MSU) crystal volumes ≥1 cm^3^ as quantified by dual-energy computed tomography (DECT) of the knees and feet/ankles (*n* = 91).

	MSU Volume < 1 cm^3^ (*n* = 55)	MSU Volume ≥1 cm^3^ (*n* = 36)	*p*-Value
Age (years)	61.4 ± 16.8	71.2 ± 11.7	**0.006**
Gout duration (years)	4.9 ± 6.6	11.2 ± 12.1	**0.002**
Gout duration >2 years	26 (47.3%)	28 (77.8%)	**0.007**
Male gender	47 (85.5%)	29 (80.6%)	0.74
BMI (kg/m^2^)	29.3 ± 4.6	29 ± 5.8	0.38
Number of flares (in last six months)	2.1 ± 2.3	2.3 ± 2.1	0.35
Tophi	6 (11.1%)	12 (33.3%)	**0.021**
MSU DECT volume (cm^3^)	0.27 ± 0.25	8.4 ± 17.2	**<0.0001**
Anti-inflammatory drugs	2 (3.7%)	2 (5.6%)	1
Serum urate level (mg/dL)	8.9 ± 1.9	9.5 ± 2.5	0.50
eGFR ≥ 60 (mL/min/1.73 m^2^)	39 (72.2%)	20 (57.1%)	0.21
Urolithiasis	8 (14.5%)	5 (13.9%)	1
Hypertension	30 (55.6%)	33 (91.7%)	**0.0006**
Stroke	4 (7.5%)	0 (0%)	0.14
Myocardial infarction	7 (13.2%)	9 (25%)	0.25
Diabetes mellitus	12 (22.6%)	14 (38.9%)	0.16
Chronic heart failure	5 (9.4%)	13 (36.1%)	**0.005**
Diuretics	12 (22.6%)	15 (41.7%)	0.09
Lipid-lowering drugs	20 (40%)	15 (44.1%)	0.88
Triglycerides level (mg/dL)	224 ± 448	154 ± 59	0.86
Cholesterol level (mg/dL)	184 ± 65	172 ± 36	0.71
LDL level (mg/dL)	103 ± 41	96 ± 35	0.50
HDL level (mg/dL)	43 ± 11	44 ± 15	0.84

BMI: body mass index; eGFR: estimated glomerular filtration rate (CKD-EPI); LDL: low-density lipoproteins; HDL: high-density lipoproteins. Bold: significant *p*-values (*p* < 0.05).

**Table 6 jcm-09-01295-t006:** Factors included in the “reduced” multiple linear regression model and associated with dual-energy computed tomography (DECT) volumes of monosodium urate (MSU) crystal deposition ≥1 cm^3^.

	Model Coefficient	OR	95% CI	*p*-Value
Gout symptom duration (per 10-year increase)	1.15	3.15	(1.60; 7.63)	0.004
Diabetes mellitus	1.56	4.75	(1.58; 15.63)	0.007
Chronic heart failure	2.06	7.82	(2.29; 31.38)	0.002

OR: Odds ratio; CI: confidence interval; MSU: monosodium urate.

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
