# Peer review of "Association of Specific Comorbidities with Monosodium Urate Crystal Deposition in Urate-Lowering Therapy-Naive Gout Patients: A Cross-Sectional Dual-Energy Computed Tomography Study"

_jcm, 2020, doi:10.3390/jcm9051295_

Round 1
Reviewer 1 Report
This is a very well-done study and I am highly impressed with it. The question is very clinically relevant. The analysis is well-done and very well explained. The results are nicely presented and the discussion is informative.
My comments are minor:
-Would it be possible that in some patients, most of the tophaceous burden was found in an area other than the knees? We can all agree in tat the feet are a natural choice, but I wonder if the ankles should have followed as a secondary site instead of the knees. Unless the authors had a reference or information about the knees being a more common site of deposition. As the DECT protocol is not fully describes (sent to another paper) we do not know how high the feet DECT went and if it included the ankles.
-I was surprised to see that, in patients naïve to ULT, serum urate had such a poor bivariate correlation with tophaceous burden. Probably this deserves a few lines in the discussion. Is a natural hypothesis to have patients with higher intercritical serum urate having higher tophaceous burden. Any explanation why this would not be the case?
-Introduction, line 58: would prefer “which factors are associated” rather than “correlated”
Author Response
Dear Editor,
We thank the reviewers for their comments and critical review of our manuscript. Please find our point-by-point answers (A) below. We hope the revised manuscript will now be suitable for publication in the Journal of Clinical Medicine.
Sincerely yours,
Tristan Pascart on behalf of all co-authors.
This is a very well-done study and I am highly impressed with it. The question is very clinically relevant. The analysis is well-done and very well explained. The results are nicely presented and the discussion is informative.
- We thank the reviewer for her/his kind comments.
My comments are minor:
-Would it be possible that in some patients, most of the tophaceous burden was found in an area other than the knees? We can all agree in tat the feet are a natural choice, but I wonder if the ankles should have followed as a secondary site instead of the knees. Unless the authors had a reference or information about the knees being a more common site of deposition. As the DECT protocol is not fully describes (sent to another paper) we do not know how high the feet DECT went and if it included the ankles.
(A) We fully agree with the reviewer’s point and indeed the feet scans included the ankles as well. We added this precision throughout the manuscript and abstract.
-I was surprised to see that, in patients naïve to ULT, serum urate had such a poor bivariate correlation with tophaceous burden. Probably this deserves a few lines in the discussion. Is a natural hypothesis to have patients with higher intercritical serum urate having higher tophaceous burden. Any explanation why this would not be the case?
(A) We thank the reviewer for this suggestion. We were also surprised to see that serum urate levels seemed of little importance to predict the extent of the tophaceous burden, and that some with high serum urate levels might not have deposited MSU crystals as much as other with however lower hyperuricemia. Our hypothesis is that as long as serum urate levels are above urate saturation levels, the kinetics of MSU crystallization are guided by comorbidities and time. This point was added in the discussion section of the revised manuscript page 8 lines 186 -191.
-Introduction, line 58: would prefer “which factors are associated” rather than “correlated”
(A) We thank the reviewer for the correction and made the change in the revised manuscript.
Reviewer 2 Report
This study examines comorbidities that are known or suspected to be associated with gout and their correlations to monosodium urate depositions evaluated by dual-energy CT in persons with gout naive to urate lowering therapy.
This study explores crystal deposition quantitatively, by splitting the dataset into deposits that are clinically meaningful opposed to those who are not.
I find the study interesting and relevant. Please address comments to improve the manuscript:
Title
Consider rephrasing including what type of study is conducted.
(ie, according to the EQUATOR network, research design [cross-sectional study?] should preferably be part of the title)
Abstract
Conclusion: The conclusion seems to conclude on secondary findings that occurred after splitting the dataset. The approach to split the dataset is relevant (exploratory) and the finding is interesting, but, conclusion should be to address the primary analyses (objective). In addition, I don’t think data support the conclusion “diabetes mellitus are more strongly associated with increased MSU crystal deposition in knees and feet than gout duration,…”.
Introduction
Did you have any hypothesis a priori of what you would expect to find? Please describe.
L. 52. What is the study specific spatial resolution both inplane and cross-sectional?
Methods
Was the study done in agreement to any protocol? Please describe and disclose an appendix if protocol exists.
(Protocol or an SAP as an PDF appendix with the paper [if accepted] would be highly valuable in order to understand to statistical context).
Setting
- Please elaborate on dates in agreement with STROBE guidelines for a cross-sectional study: “Describe the setting, locations, and relevant dates, including periods of recruitment, exposure, follow-up, and data collection.”
Analysis population
- Were any calculations carried out prior to initiation of the study to decide how many participants to include? Please describe.
- It appears from the results section that the per protocol population was chosen for analysis. When was this decided – any statistical protocol? Any sensitivity assessing best/worst/multiple imputation? For serum-lipids analysis over 1/3 of the ITT population is left out of analysis due to missing data, you mention this a limitation in L. 209-210.
- Please state the analysis population clearly in the methods section and justify why you did not use the ITT population.
- Please state clearly how missing data are handled.
L.100 flowchart does not add up. 125 – 18 – 12 = 95. But only 91 are included for analysis. Why are 4 additional participants excluded? Please update flowchart with explanation.
L.84 + 91: “Multivariate analysis was then applied to search for factors affecting MSU volumes using multiple linear regression models integrating variables with p-values < 0.2 in bivariate analysis.” Why was a cut-off of p<0.2 chosen?
Results
Through-out the results section there is a problem with the descriptive statistics were a mean ± SD are given, and the sd exceeds the mean. For all mean ± SD please check that numbers are correct (I suspect they may be switched in some places), and if indeed SD exceeds the mean, consider substituting means ± SD with median and percentiles. To avoid confusion, please state clearly in the text which value is which, e.g. mean xxx ± SD xxx.
L. 104 table 1:
- For gout duration, number of flares and triglyceride levels the SD exceeds the mean. See comment above.
- Why are some patient characteristics lacking from the population (e.g. one participant does not have data on hypertension)?
- For "skewed data" it might be better to use Median and Interquartile Range (min;max)
L.108-113. Again, the SD exceeds the mean by far. See comment above.
L. 123-124. Table 3. Same problem as previous with means and SD. See comment above.
L. 150 Table 6 – how was factors for the reduced model chosen? Please describe. Why not include tophi which appears to have significantly different incidence between MSU <1 cm^3 and MSU >1 cm^3 (table 5)?
L. 143-144. Same problem as previously with SD exceeding mean for gout duration, number of flares, DECT MSU volume and triglyceride levels. See comment above.
L. 157 Could you please provide a small text to aid interpretation of this figure, specifying what it is the model predicts.
Serum urate levels are explored as a continuous variable. From table 1 it is evident that mean serum urate is above normal range (for both genders), but looking at SD, some participants may have levels within normal range. It is difficult to deduce from a mean continuous variable, how many participants actually have hyperuricemia, especially because the cut-off differs between genders. Consider exploring serum urate as a binary variable (above normal: yes/no), both as a participant characteristic in table 1 and for analysis like the bivariate analysis in table 3.
Discussion
L. 73-75 The study uses of two different DECT scanners. From the references I gather following information.
- Toshiba 80/135 kVp, 0.5 mm slices thickness. MSU gradient 1.07
- Siemens 80/140 kV, 1 mm slide thickness. MSU: HU threshold 150, material gradient 1.25
The scanners differ in scan protocol (tube voltages) and slice thickness. Regarding image analysis, one scanner deals with an attenuation cut-off at 150 HU, at the other does not. Please include a brief discussion on whether and how use of different dual-energy CT scanners could/may have impacted the results.
L.131-138 – Table 4 presents that the only variable significantly associated with DECT MSU volume is gout duration. However, when splitting the data into MSU < and > 1 cm^3, it appears that diabetes mellitus and chronic heart failure are associated with MSU > 1 cm^3. I find it challenging to truly understand the finding from table 4, especially in relation to the later findings. Consider adding more explanatory text in the results section or discussing it in the discussion.
Author Response
This study examines comorbidities that are known or suspected to be associated with gout and their correlations to monosodium urate depositions evaluated by dual-energy CT in persons with gout naive to urate lowering therapy.
This study explores crystal deposition quantitatively, by splitting the dataset into deposits that are clinically meaningful opposed to those who are not.
I find the study interesting and relevant.
(A) We thank the reviewer for her/his kind comments and thorough review of our manuscript.
Please address comments to improve the manuscript:
Title
Consider rephrasing including what type of study is conducted.
(ie, according to the EQUATOR network, research design [cross-sectional study?] should preferably be part of the title)
(A) We thank the reviewer for the suggestion and modified the title accordingly by adding the term “cross-sectional”.
Abstract
Conclusion: The conclusion seems to conclude on secondary findings that occurred after splitting the dataset. The approach to split the dataset is relevant (exploratory) and the finding is interesting, but, conclusion should be to address the primary analyses (objective). In addition, I don’t think data support the conclusion “diabetes mellitus are more strongly associated with increased MSU crystal deposition in knees and feet than gout duration,…”.
(A) We understand the reviewer’s point but we kindly disagree that these are secondary findings. Indeed, the linear regression analysis and the subsequent bivariate analysis splitting the dataset according to a specific cut-off were both primary analyses, “two sides of the same coin”.
Introduction
Did you have any hypothesis a priori of what you would expect to find? Please describe.
(A) Given the overall lack of previous data on this research question (“which factors are associated with the volume of monosodium urate crystal deposition quantified by dual-energy CT in urate lowering therapy-naive gout patients”) as explained in the Introduction section, the study was honestly exploratory/observational, without a priori hypothesis.
L. 52. What is the study specific spatial resolution both inplane and cross-sectional?
(A) In the three different centers, DECT images were reconstructed at a section thickness/interval of approximately 0.6/0.3 mm, yielding almost isotropic voxels of approximately 0.6 × 0.6 × 0.6 mm. We have added some details on DECT protocols to the Methods section (page 2 lines 78-79).
Methods
Was the study done in agreement to any protocol? Please describe and disclose an appendix if protocol exists.
(Protocol or an SAP as an PDF appendix with the paper [if accepted] would be highly valuable in order to understand to statistical context).
(A) The study protocol used is summarized in the Methods section. There is unfortunately no SOP or relevant additional details to provide as an Appendix and the institutional ethics committee approval is in French.
Setting
- Please elaborate on dates in agreement with STROBE guidelines for a cross-sectional study: “Describe the setting, locations, and relevant dates, including periods of recruitment, exposure, follow-up, and data collection.”
(A) We thank the reviewer for the suggestion and added the locations and periods of recruitment in the Methods section of the revised manuscript. (page 2 lines 64-65)
Analysis population
- Were any calculations carried out prior to initiation of the study to decide how many participants to include? Please describe
(A) As mentioned above, given the lack of previous data on this research topic, the required number of participants could not be determined a priori. All ULT-naive gout patients who had been assessed in the three involved centers so far were therefore included in this study.
- It appears from the results section that the per protocol population was chosen for analysis. When was this decided – any statistical protocol? Any sensitivity assessing best/worst/multiple imputation? For serum-lipids analysis over 1/3 of the ITT population is left out of analysis due to missing data, you mention this a limitation in L. 209-210.
(A) Given that this is not a comparative nor a longitudinal study trial, per protocol or ITT analyses do not apply. Indeed, as stated in the Methods and Limitations sections, over a third of the population was not included in the serum-lipids analysis.
- Please state the analysis population clearly in the methods section and justify why you did not use the ITT population.
(A) Same as above, ITT or PP analyses do not apply for this type of study.
- Please state clearly how missing data are handled.
(A) No specific imputations were performed where there were missing data. This was clarified in the Methods section (page 2 line 73).
L.100 flowchart does not add up. 125 – 18 – 12 = 95. But only 91 are included for analysis. Why are 4 additional participants excluded? Please update flowchart with explanation.
(A) We thank the reviewer for pointing out this error, there were in fact 4 additional patients with missing knee DECT scans. This has now been corrected in the revised flow diagram.
L.84 + 91: “Multivariate analysis was then applied to search for factors affecting MSU volumes using multiple linear regression models integrating variables with p-values < 0.2 in bivariate analysis.” Why was a cut-off of p<0.2 chosen?
(A) The cut-off of p<0.2 was specifically chosen by our biostatistician (co-author) to avoid missing variables which could become statistically significant when adjusted on the other factors in the multivariate analysis (rather than taking variables with p<0.05 in bivariate analysis).
Results
Through-out the results section there is a problem with the descriptive statistics were a mean ± SD are given, and the sd exceeds the mean. For all mean ± SD please check that numbers are correct (I suspect they may be switched in some places), and if indeed SD exceeds the mean, consider substituting means ± SD with median and percentiles. To avoid confusion, please state clearly in the text which value is which, e.g. mean xxx ± SD xxx.
(A) We thank the reviewer for the suggestion and added a comment in the Methods section (page 2 lines 87-88) and added throughout the revised text the medians and interquartile ranges, where appropriate. We also double-checked the means and SD numbers and they are correct.
L. 104 table 1:
- For gout duration, number of flares and triglyceride levels the SD exceeds the mean. See comment above.
(A) The medians and interquartile ranges were added in Table 1 for these variables following this comment.
- Why are some patient characteristics lacking from the population (e.g. one participant does not have data on hypertension)?
(A) These patient characteristics were not reported by the investigator on the CRF and were therefore unavailable for the analysis.
- For "skewed data" it might be better to use Median and Interquartile Range (min;max)
(A) As requested, medians and IQRs have been added where appropriate throughout the revised manuscript.
L.108-113. Again, the SD exceeds the mean by far. See comment above.
(A) As requested, we have added medians and IQRs for the main values in these lines.
L. 123-124. Table 3. Same problem as previous with means and SD. See comment above.
(A) We understand the reviewer’s point, however there is already a large amount of data presented in Table 3, therefore no additions were made in for the sake of clarity.
L. 150 Table 6 – how was factors for the reduced model chosen? Please describe. Why not include tophi which appears to have significantly different incidence between MSU <1 cm^3 and MSU >1 cm^3 (table 5)?
(A) We thank the reviewer for this comment. The statistical software and model work by automatically selecting relevant items, and subcutaneous tophi were automatically removed by the model itself. We added this explanation for subcutaneous tophi in the Results section (page 8 line 166).
L. 143-144. Same problem as previously with SD exceeding mean for gout duration, number of flares, DECT MSU volume and triglyceride levels. See comment above.
(A) Same answer as for Table 3, there is already a large amount of data in the Table, and adding the medians and IQRs would not change dramatically the understanding for the reader but rather create an unnecessary ‘heavy’ Table.
L. 157 Could you please provide a small text to aid interpretation of this figure, specifying what it is the model predicts.
(A) We thank the reviewer for this suggestion and completed the interpretation of Figure 2 on page 8 lines 167-169.
Serum urate levels are explored as a continuous variable. From table 1 it is evident that mean serum urate is above normal range (for both genders), but looking at SD, some participants may have levels within normal range. It is difficult to deduce from a mean continuous variable, how many participants actually have hyperuricemia, especially because the cut-off differs between genders. Consider exploring serum urate as a binary variable (above normal: yes/no), both as a participant characteristic in table 1 and for analysis like the bivariate analysis in table 3.
(A) We believe the reviewer is did not understand this part of the study, as all of our patients were untreated (“ULT-naive”) gout patients and thus had serum urate levels above 6.0mg/dL, which is now the universally recognized threshold of hyperuricemia for both genders (the distinction between genders does not exist anymore). As a matter of fact, the lowest value of serum urate was 6.9 mg/dL, still above 6.0mg/dL.
Discussion
L. 73-75 The study uses of two different DECT scanners. From the references I gather following information.
- Toshiba 80/135 kVp, 0.5 mm slices thickness. MSU gradient 1.07
- Siemens 80/140 kV, 1 mm slide thickness. MSU: HU threshold 150, material gradient 1.25
The scanners differ in scan protocol (tube voltages) and slice thickness. Regarding image analysis, one scanner deals with an attenuation cut-off at 150 HU, at the other does not. Please include a brief discussion on whether and how use of different dual-energy CT scanners could/may have impacted the results.
(A) To our knowledge, comparative studies on the diagnostic accuracy of different DECT scanners in gout are not yet available. In the case of our study, having used two different DECT manufacturers should not significantly affect the results and conclusions given that the vast majority of DECT scanners were performed using one vendor (Siemens) while less than 10% of DECT scanners were performed using the second manufacturer (Canon). Also, for all DECT scanners involved in this study, we have used the default gout dual-energy post-processing protocol, which has been used and reported in several publications with these two DECT manufacturers so far. We have added a comment on this in the Methods and Limitations section (page 9 lines 237-242).
L.131-138 – Table 4 presents that the only variable significantly associated with DECT MSU volume is gout duration. However, when splitting the data into MSU < and > 1 cm^3, it appears that diabetes mellitus and chronic heart failure are associated with MSU > 1 cm^3. I find it challenging to truly understand the finding from table 4, especially in relation to the later findings. Consider adding more explanatory text in the results section or discussing it in the discussion.
(A) We understand the point of the reviewer. Actually, both analyses provide the same results but the level of significance is not the same. In the linear regression analysis, there is a clear trend identifying CHF as a determining factor to explain increasing DECT volumes of MSU crystal deposition with a similar coefficient as subcutaneous tophi. However, this finding did not reach statistical significance because the 95% confidence interval included 0. The most probable explanation is that the study is a little underpowered for this specific analysis given that subcutaneous tophi also just included 0 in the 95% confidence interval (and by definition they are truly associated with higher MSU crystal volumes). A comment was added in the Results (page 7 lines 148-149).
Reviewer 3 Report
The paper by Pascart et al describes which factors are associated with the density of MSU crystal deposition as quantified by the relatively new imaging technique DECT in the context of gout. A total of 125 patients who had not used therapy for their gout were included and the data of 91 patients could be used. After multiple analyses (with some missing data) the authors can conclude that high concentration of crystal deposition on DECT scan was associated with gout duration, diabetes and chronic heart failure.
This research group is well established in gout and the associated use of DECT. In the clinical field a lot of knowledge on DECT scans is still missing so their study is important. Study limitations are well assessed in the discussion and overall the paper is very well written.
I only have some minor points.
Minor
- It’s unclear how data can be missing if patients were included consecutively and in a prospective fashion. How is this possible?
- Different imaging machines from different companies were used to assess crystal deposition: is this a problem?
- References: please truncate the number of authors from 3 onwards, after 4 or more authors.
Author Response
The paper by Pascart et al describes which factors are associated with the density of MSU crystal deposition as quantified by the relatively new imaging technique DECT in the context of gout. A total of 125 patients who had not used therapy for their gout were included and the data of 91 patients could be used. After multiple analyses (with some missing data) the authors can conclude that high concentration of crystal deposition on DECT scan was associated with gout duration, diabetes and chronic heart failure.
This research group is well established in gout and the associated use of DECT. In the clinical field a lot of knowledge on DECT scans is still missing so their study is important. Study limitations are well assessed in the discussion and overall the paper is very well written.
- We thank the reviewer for her/his kind comments
I only have some minor points.
Minor
- It’s unclear how data can be missing if patients were included consecutively and in a prospective fashion. How is this possible?
- We thank the reviewer for this comment. As the reviewer may have noticed, there is significant missing data almost exclusively for lipid levels because these were not mandatory in the initial assessment. We added the precision in the Methods section page 2 line 71.
- Different imaging machines from different companies were used to assess crystal deposition: is this a problem?
- This is a point that has not been addressed in the literature so far, however all of these machines are used in clinical practice and we used the volumes that are rendered daily to physicians. They are not expected to be source of major discrepancies.
- References: please truncate the number of authors from 3 onwards, after 4 or more authors.
- We thank the reviewer for the recommendation and made the changes accordingly.
Round 2
Reviewer 2 Report
Thank you addressing concerns. I still believe the manuscript could improve with minor changes in the methodology and statistics.
- Abstract: Thank you for the reply. I still don’t think data support the conclusion “diabetes mellitus are more strongly associated with increased MSU crystal deposition in knees and feet than gout duration,…”. I find the words “more strongly” misleading: When looking at CI’s for gout duration and DM, the true value of the OR could be identical, or it could even be slightly lower for DM.
- I appreciate that the method section reflects the initial protocol. For transparency, it would be suitable to make the original protocol an appendix or make it public available, regardless of language. Please consider if this is possible.
- “given the lack of previous data on this research topic, the required number of participants could not be determined a priori. All ULT-naive gout patients who had been assessed in the three involved centers so far were therefore included in this study.” Please describe how you decide when to stop inclusion, if not by the number of patients.
- I found the following answer very informative, please consider including it the manuscript: “The cut-off of p<0.2 was specifically chosen by our biostatistician (co-author) to avoid missing variables which could become statistically significant when adjusted on the other factors in the multivariate analysis (rather than taking variables with p<0.05 in bivariate analysis).”
- In some places, the SD still exceeds the mean. You have stated this as a deliberate choice due to large amount of data and for the sake of clarity. To me, it is not a matter of clarity but a matter of presenting statistical meaningful data. When SD exceeds the mean, and the value represented is a volume, reading this, it means that the true value of the volume could in fact be negative. This is obviously not the case. To minimize the amount of data, one could choose to simply substitute with more appropriate descriptive statistics.
- “The statistical software and model work by automatically selecting relevant items, and subcutaneous tophi were automatically removed by the model itself. We added this explanation for subcutaneous tophi in the Results section (page 8 line 166).” Please describe the statistical model (forward/backward selection or other method?) used for selection.
- Thank you for your answer regarding serum urate levels. I was not aware of the universally recognized threshold but compared to national reference standard – my apologies. You may once again consider replacing the mean and sd with median and range. Currently you state the mean to be 9.1 ±2.2, as a reader, I read 95% of the values to lie approximately between 4.7 and 13.5. From the point-to-point answer the lowest value is 6.9. The descriptive statistics does not reflect this very well.
Author Response
Dear Editor,
We thank the reviewer for her/his comments and critical review of our manuscript. Please find our point-by-point answers (A) below. We hope the revised manuscript will now be suitable for publication in the Journal of Clinical Medicine.
Sincerely yours,
Tristan Pascart on behalf of all co-authors.
Thank you addressing concerns. I still believe the manuscript could improve with minor changes in the methodology and statistics.
Abstract: Thank you for the reply. I still don’t think data support the conclusion “diabetes mellitus are more strongly associated with increased MSU crystal deposition in knees and feet than gout duration,…”. I find the words “more strongly” misleading: When looking at CI’s for gout duration and DM, the true value of the OR could be identical, or it could even be slightly lower for DM.
- We understand the reviewer’s point, but strictly speaking the OR of both comorbidities are higher than the one of 10-year gout duration (which is a very high disease duration for patients untreated for all this time), therefore the probability is significantly higher that the association is indeed stronger for comorbidities than disease duration (even when we consider very long untreated disease duration). We therefore kept the abstract as it is.
I appreciate that the method section reflects the initial protocol. For transparency, it would be suitable to make the original protocol an appendix or make it public available, regardless of language. Please consider if this is possible.
- We thank the reviewer for the suggestion, but unfortunately the original protocol of the cohort deals with several scientific questions, and would not add anything to what is already stated in the methods section as per our .R1 reply.
“given the lack of previous data on this research topic, the required number of participants could not be determined a priori. All ULT-naive gout patients who had been assessed in the three involved centers so far were therefore included in this study.” Please describe how you decide when to stop inclusion, if not by the number of patients.
- Inclusions in this prospective cohort are still on-going but we decided to freeze the data when we wanted to address this scientific question, and 91 patients seemed sufficient.
I found the following answer very informative, please consider including it the manuscript: “The cut-off of p<0.2 was specifically chosen by our biostatistician (co-author) to avoid missing variables which could become statistically significant when adjusted on the other factors in the multivariate analysis (rather than taking variables with p<0.05 in bivariate analysis).”
- We thank the reviewer for the suggestion and added the sentence in the Methods section (page 3 lines 100-102).
In some places, the SD still exceeds the mean. You have stated this as a deliberate choice due to large amount of data and for the sake of clarity. To me, it is not a matter of clarity but a matter of presenting statistical meaningful data. When SD exceeds the mean, and the value represented is a volume, reading this, it means that the true value of the volume could in fact be negative. This is obviously not the case. To minimize the amount of data, one could choose to simply substitute with more appropriate descriptive statistics.
- We thank the reviewer for her/his suggestion and replaced all means in Table 3 by medians and IQR which dealt with volumes.
“The statistical software and model work by automatically selecting relevant items, and subcutaneous tophi were automatically removed by the model itself. We added this explanation for subcutaneous tophi in the Results section (page 8 line 166).” Please describe the statistical model (forward/backward selection or other method?) used for selection.
- The statistical model is described in the Methods page 3 line 103: “The step-by-step backward method, based on Akaike information criterion, was selected for the reduced model.”
Thank you for your answer regarding serum urate levels. I was not aware of the universally recognized threshold but compared to national reference standard – my apologies. You may once again consider replacing the mean and sd with median and range. Currently you state the mean to be 9.1 ±2.2, as a reader, I read 95% of the values to lie approximately between 4.7 and 13.5. From the point-to-point answer the lowest value is 6.9. The descriptive statistics does not reflect this very well.
- We thank the reviewer for the suggestion and added the medians and IQR of serum urate levels in table 1.